# Linguistic correlates of societal variation: A quantitative analysis

**Sihan Chen**[1☯], **David Gil**[2☯], **Sergey Gaponov**[3], **Jana Reifegerste**[4], **Tessa Yuditha**[5], **Tatiana Tatarinova**[3], **Ljiljana Progovac**[6‡], **Antonio Benítez-Burraco**[5‡]*

1 Department of Brain and Cognitive Sciences, Massachusetts Institute of Technology, Cambridge, MA, United States of America, 2 Department of Linguistic and Cultural Evolution, Max Planck Institute for Evolutionary Anthropology, Leipzig, Germany, 3 Department of Biology and Computational Biology, University of LaVerne, LaVerne, CA, United States of America, 4 Department of Neurology, Georgetown University, Washington, DC, United States of America, 5 Department of Spanish, Linguistics & Theory of Literature, University of Seville, Seville, Spain, 6 Linguistics Program, Wayne State University, Detroit, MI, United States of America

☯ These authors contributed equally to this work.
‡ LP and ABB also contributed equally to this work.
* abenitez8@us.es

**Data Availability Statement:** The data generated by this research and used in the analyses can be found at https://github.com/cshnican/XSlanguages

**Funding:** This research was supported by grant PID2020-114516GB-I00 funded by MCIN/AEI/

## Abstract

Traditionally, many researchers have supported a uniformitarian view whereby all languages are of roughly equal complexity, facilitated by internal trade-offs between complexity at different levels, such as morphology and syntax. The extent to which the speakers' societies influence the trade-offs has not been well studied. In this paper, we focus on morphology and syntax, and report significant correlations between specific linguistic and societal features, in particular those relating to exoteric (open) vs. esoteric (close-knit) society types, characterizable in terms of population size, mobility, communication across distances, etc. We conduct an exhaustive quantitative analysis drawing upon WALS, D-Place, Ethnologue and Glottolog, finding some support for our hypothesis that languages spoken by exoteric societies tend towards more complex syntaxes, while languages spoken by esoteric societies tend towards more complex morphologies.

## 1. Introduction

For many years, the uniformitarian view of languages has claimed that all languages are roughly equal in terms of their overall complexity [1, 2]. This equi-complexity of languages has been further hypothesized to entail a trade-off principle, in accordance with which, if one language exhibits a more complex morphology, it will have a simpler syntax, so that their overall complexity will be the same [3, 4]. Moreover, it is commonly assumed that these trade-offs are mostly internally-motivated, with factors external to language, like sociopolitical characteristics, cultural traits, or the physical environment, playing minor roles, if any, in shaping language features and language diversity. At most, the effects of these factors have been circumscribed to quite peripheral components of language, particularly, the lexicon. To a great extent, this uniformitarian view of languages results from a uniformitarian view of the

10.13039/501100011033 (to ABB). The funders had no role in study design, data collection and analysis, decision to publish, or preparation of the manuscript.

**Competing interests:** The authors have declared that no competing interests exist.

cognitive faculty that makes it possible to learn and use languages (i.e. our faculty of language, language-ready brain, or human linguisticality), which has been assumed by some to be the same in all human beings and to have remained unmodified since our inception as a species [5, 6]. The Chomskyan approach to language evolution and language diversity nicely exemplifies this view [7].

Increasing evidence suggests, however, that overall language complexity might differ cross-linguistically [8–10]. Additionally, research suggests that trade-offs, within specific domains or across diverse domains, might not necessarily entail equal overall complexity [11–13]. Some research has even cast doubt on the existence of such trade-offs [14–16], in particular between morphology and syntax [17]. Lastly, specific language features have been shown to be impacted by extralinguistic factors. In particular, phonological features of languages might adapt to the physical environment in which they are spoken. A familiar example is the effect of vegetation on sound inventories, with the languages spoken in tree-covered areas showing a greater proportion of vowels [18], which parallels what has been observed in many vertebrates [19, 20]. Another example is the negative effect of dry climates on tone usage: the global distribution of tonal languages, which are concentrated in tropical and subtropical regions, is arguably explained by the suboptimal phonation caused by desiccated and cold air [21]. Likewise, changes in the human body, particularly the jaws, have been argued to affect the distribution of the sounds of world languages [22, 23], and how phonological inventories have changed over time [24]. Still, the effect of the physical environment on language features is more frequently exerted via its influence on diverse aspects of human ecology (like shortages of food supply or the spread of diseases) and human sociology (such as demographic changes, migrations and population contacts, or changes in social networks) [25]. Not surprisingly then, our social environment may have a considerable impact on the structure of languages. Recent typological surveys suggest that the number of speakers, the degree of bilingualism, the tightness or the looseness of the social networks, the sociopolitical organization, or the number of adult learners of a language correlate, and perhaps explain, the types of morphology or syntax exhibited by the world languages [26–28]. Specific examples are the negative correlation found between the morphological complexity measures and population size [29], the positive correlation between cultural/socio-political complexity and tense–aspect–mood (TAM) marking, as well as thematic-role assignment [30], or the positive correlation between population size and the complexity in core argument marking [31]. That said, the potential impact of this type of sociopolitical factors on putative trade-offs between parts of grammar has been addressed by very few (if any) works (see [32] for an attempt), hence the novelty of our study.

When one considers all the social factors with an impact on language structure together with the language features subject to variation, some interesting patterns emerge (see [33, 34] for seminal discussions). Large and complex social networks, involving greater rates of inter-group contacts and cultural exchanges (i.e., *open* or *exoteric societies*) seemingly favor languages with expanded vocabularies, greater compositionality and enhanced semantic transparency, as well as more complex and more layered syntaxes, with more specialized and obligatory grammaticalized distinctions and greater reliance on embedding. These languages also seem to exhibit less complex phonologies and morphologies. In this paper, we will call them *Type X* (from eXoteric) *languages*. By contrast, the languages spoken by isolated human groups living in small and tight communities with high proportions of native speakers (i.e., *close-knit* or *esoteric societies*) seem to exhibit larger sound inventories and more complex phonotactics, more complex and more opaque morphologies (with more irregularities and morpho-phonological constraints), reduced semantic transparency and compositionality (with an abundance of idioms and idiosyncratic constructions), as well as simpler and less layered syntaxes. In this paper, we will refer to these languages as *Type S* (from eSoteric) *languages*.

Overall, the differences between Type X and Type S languages (which are similar to the differences between the languages spoken by Type 2 and Type 1 communities, respectively, in [35]) can be associated with their differential context-dependency. Specifically, Type X languages seem to be optimized for decontextualized language uses, whereas Type S languages are used by people sharing considerable amounts of knowledge. Likewise, Type X languages might be optimized for being learned by adults, whereas Type S languages might be better learnable by children. Ultimately, this evidence suggests that language diversity can have an adaptive value, with language structures adapting to the social niches in which they are being learned and used. This is the Linguistic Niche Hypothesis [36].

In this paper, we conduct an extensive quantitative analysis of the structural diversity of the world's languages, drawing upon one comprehensive typological database, as well as of the cultural and sociopolitical diversity of world human groups, drawing upon several different sociological and cultural databases, in order to determine whether a correlation, and perhaps also causation, exists between specific linguistic and societal features, in particular, those relating to exoteric vs. esoteric society types. Specifically, we test the hypothesis that esoteric societies speak languages featuring more complex morphologies (aka Type S languages), whereas the languages spoken by exoteric societies exhibit greater complexity in syntax (aka Type X languages).

## 2. Methods

To quantify the relation between societal exotericity and language complexity, we drew data from four different, independently constructed databases. Language features were drawn from the World Atlas of Language Structures (WALS) [37]. Meanwhile, societal features were collected from three databases: Ethnologue [38], Glottolog [39], and D-Place [40].

WALS is a database containing various language features in domains including phonology, morphology, syntax, and lexical semantics; in this paper we focus on features related to morphology and syntax. Each feature permits different values numerically coded in the database. To facilitate our analysis, we construct a classification of feature values, drawn from 82 of the 142 language features covered in WALS, with each feature described and visualized in its own chapter. For example, Chapter 26 of WALS concerns the affixing in inflectional morphology and contains 6 different feature values: 1 (little or no inflectional morphology), 2 (predominantly suffixing), 3 (moderate preference for suffixing), 4 (approximately equal amounts of suffixing and prefixing), 5 (moderate preference for prefixing), and 6 (predominantly prefixing). A potential classification of these features could be 1<2/3/4/5/6, which separates languages with little or no inflectional morphology from those with some degree of inflectional morphology. We then say that the latter category is more complex than the former one, following the principle that the more symbols needed to fully describe a grammatical rule, the more complex the rule is [41]. In this case, more text is needed to describe a grammar with affixes than to describe a grammar without them, since to describe the former, one needs to specify explicitly the forms, functions and locations of the affixes, whereas no description is needed for the latter. In this paper, we code this classification as "Existence of affixes (no < yes)", where languages with affixes (the "yes" category) are considered more complex than those without (the "no" category). On the other hand, there could be more than one classification of feature values within the same feature in WALS. For example, WALS Chapter 30 pertains to the number of grammatical genders across languages, ranging from 1 (no grammatical genders) to 5 (five or more grammatical genders). We can have two classifications in this case: one related to the existence of grammatical genders, in which case, using the aforementioned conventions, the classification would be 1<2/3/4/5, and another related to the number of

grammatical genders, in which case the classification would be 1<2<3<4<5 (henceforth sim-plified as 1<<5). In total, from the 82 WALS features we constructed 94 feature classifications. We then considered whether each feature classification is related to morphology or syntax. Recognizing that demarcating morphological features from syntactic ones is an active debate in linguistics [42–45], *inter alia*), here we adopted a simple criterion that if a feature classifica-tion is related to grammatical rules within a word, then it is considered as a morphological classification; in addition, if a feature classification relates to grammatical rules between words, then we consider it as a syntactic classification. For example, the classification of number of grammatical genders is considered morphological, since most languages distinguish grammat-ical genders through morphological markers, whereas the existence of a dominant word order is considered syntactic, since it concerns the order among words. Still, as noted, some features can be assigned to both domains. For example, the classification of number of cases can be considered morphological, since cases involve changing the word form through different inflectional endings. However, cases are used to mark sentence constituents and relationships between phrases, hence they play a role as well at the sentence level. Conversely, passive con-structions mostly involve changing the sentence structure, but they are usually marked through specific affixes in the verb, so passives also have a morphological dimension. Accord-ingly, in our analysis we adopted a quadripartite criterion, distinguishing between purely mor-phological features (M), purely syntactic features (S), features pertaining to both domains but predominantly related to morphology (Ms) and features pertaining to both domains but pre-dominantly related to syntax (mS) (see Fig 2 for details).

The first two societal features we considered pertain to the current status of the language within its society. This is quantified by the Expanded Graded Intergenerational Disruption Scale (EGIDS) published by Ethnologue. A language can be assigned one of the 13 values between 0 and 10 (there are two values of 6 and two values of 8, each suffixed by "a" and "b", respectively). A value of 0 indicates that the language is widely used internationally in a broad range of activities, whereas a value of 10 indicates that the language is no longer used. There-fore, we consider an EGIDS value of 0 to be an extreme case of exotericity and a 10 as an extreme case of esotericity. In our study, we adopted two scales for language status. The first scale (henceforth referred to as EGIDS) reflects the gradient nature of EGIDS, where the value 1 corresponds to the original EGIDS value 10 (extinct), and the value 13 corresponds to the original value of 0 (international language). The original values of 8b, 8a, 6b, and 6a corre-spond to 3, 4, 6, and 7 in our scale, respectively. The second scale (labeled as EGIDSnat) is whether a language is a national language or not, with 1 indicating the language is not a national language, and 2 indicating the language is a national language.

The next societal feature is the size of the language family that a language belongs to, quanti-fied by the number of languages belonging to the same language family, according to the Glot-tolog classification. The family sizes range from 1 (individual language isolates) to 1433 (languages belonging to the Atlantic-Congo family). Societies where people speak a language belonging to a larger family tend to be more exoteric, since they are more likely to be a result of previous rapid expansion and migration, often due to technological development [30]. Con-versely, societies where people speak a language belonging to a smaller language family tend to be more esoteric.

In addition, we drew 6 features from the D-Place database measuring the degree of com-plexity of a society, including the number of jurisdictional levels above the local community (Feature EA033 in the database), the size of local communities (EA031), population size (EA202) and density (SCCS156), fixity of residence (SCCS150), and distance moved each year (B014). An exoteric society tends to have more jurisdictional levels, larger local communities,

larger population size, and higher population density; moreover, people living in an exoteric society are also less likely to settle at a place and therefore more likely to move around.

These 9 societal features are largely correlated to each other, such as EGIDS and EGIDSnat. A potential issue of having correlated features is that they may inflate the number of significant correlations between linguistic feature classifications and sociopolitical features. To account for this, we first imputed the missing values in the dataset using the missforest package [46] in R [47]. Then, we ran a principal component analysis (PCA) on these 9 features to extract dimensions that capture the most variance in the data, using the prcomp function in R. The first principal component (PC1 henceforth) explained 56.76% of the variance in the data, suggesting that all 9 features broadly vary along the axis of exotericity and esotericity. Fig 1 shows the loading of each sociopolitical variable onto the first two PCs: the more negative a PC value is for a society, the higher complexity it has.

Bringing together the above sources, we constructed a dataset containing 94 different classifications along with 1 societal PC. We ran a linear regression between each combination of a classification and the PC, resulting in 94 statistical tests. For binary classifications, namely those with only two values, we ran a logistics regression instead. For each statistical test, we reported the estimated slope along with the p-value. We say a relation between a principal component is significant if the p-value is less than 0.05.

The method described above tests the correlations between classifications of linguistic feature values and societal features on a global scale. However, this set of tests leaves the following question unanswered: are the correlations actually driven by societal features, or alternatively, by other factors such as language family and geographical regions? To control for these potential confounds, usually referred to as Galton's problem [48], we conducted an additional analysis, taking into account the phylogeny and the geographical proximity of languages. In brief, for each combination of a classification and the PC, we ran a Bayesian mixed-effects linear (for binary classifications, logistics) regression, using the brms package [49] in R [47]. The PC values were coded as fixed effects, and we fully specified the random effects of phylogeny and geographical proximity by two covariance matrices. The covariance matrix for phylogeny is obtained from a reconstructed global phylogeny tree [50] using the ape package [51] in R [47]. Two languages have higher covariance if they're closely located on the phylogenetic tree (e.g. English and Dutch) and lower covariance if they're not (e.g. Turkish and Guarani). The covariance matrix for geography is based on the spatial distance of each 2 languages calculated from the coordinates provided in WALS [37], using the geoR package [52]. The distances were first transformed to Matérn covariances by the varcov.spatial function and then normalized against the maximum covariance. Following the syntax in brms, the regression (linear or logistic) equation can be written as follows:

$$
\begin{aligned}
\text{grammatical classification} \sim \text{PC} + (1|\text{gr}(\text{Glottocode, spatial covariance matrix})) \\
+ (1|\text{gr}(\text{Glottocode, phylogenetic covariance matrix}))
\end{aligned}
\tag{1}
$$

Each test generated a posterior distribution of the slope estimate. We reported the lower 2.5% quantile, the posterior mean, and the upper 97.5% quantile. A result was significant if the 2.5% quantile and 97.5% quantile were both above or below zero.

## 3. Results

Fig 2 shows the regression results of the global analysis, paneled first by whether a grammatical feature is broadly considered as morphological or syntactic and then by whether a grammatical feature predominately falls into one category but has some relations with the other. Since in our dataset, a more negative PC value indicates a higher sociopolitical complexity, in Fig 2

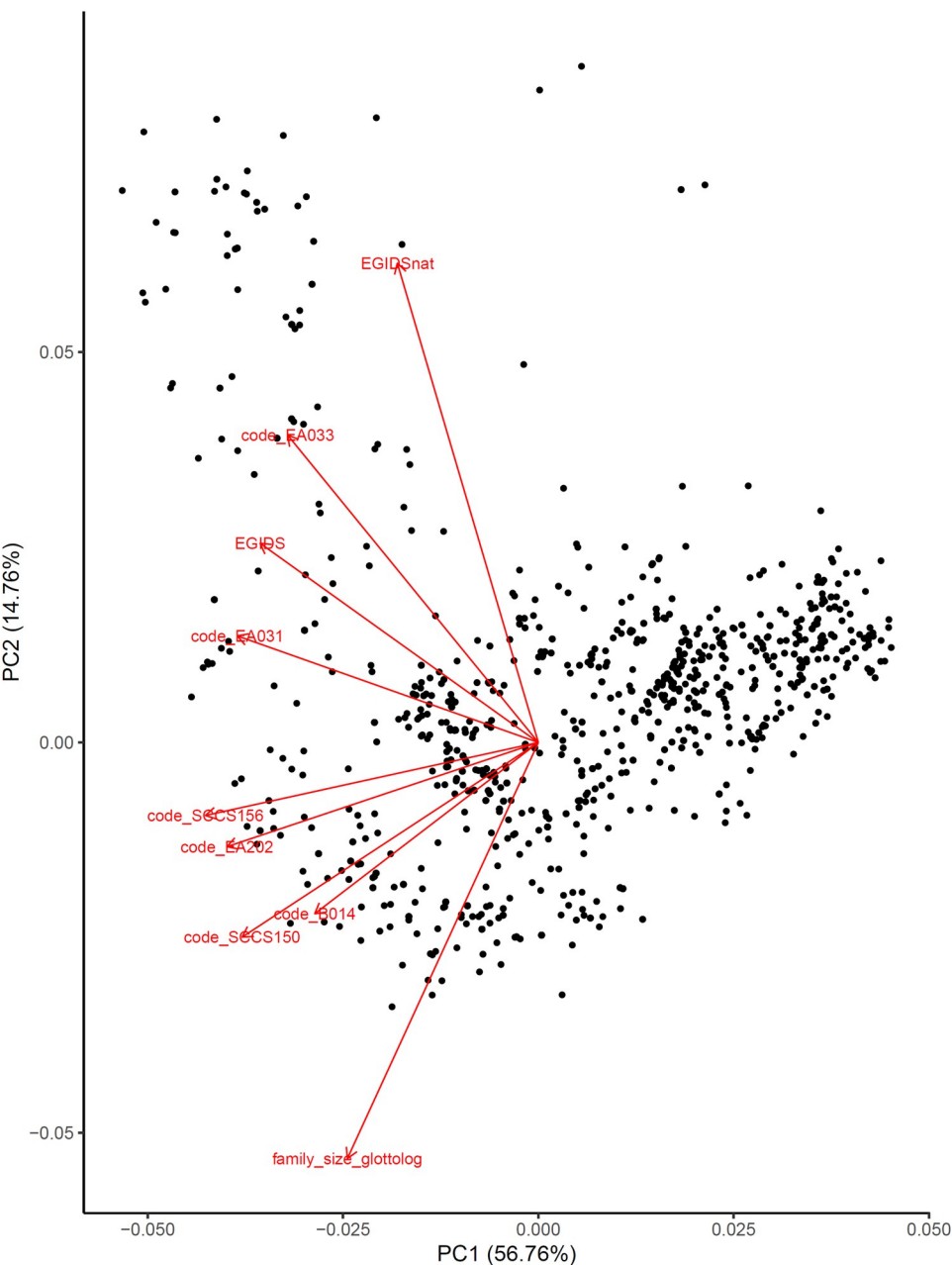

**Fig 1. The loading plot of different sociopolitical variables from the principal component analysis (PCA).** Each dot represents a society. The red arrows represent the loading of each variable onto the first two principal components (PCs). The first PC (PC1) is the one used in this study quantifying sociopolitical complexity. A more negative PC value represents a higher sociopolitical complexity.

(and similarly in Fig 3), a positive regression slope (a red dot) indicates a negative relationship between grammatical complexity and sociopolitical complexity, and similarly, a negative regression slope (a blue dot) indicates a positive relationship between grammatical complexity and sociopolitical complexity. Therefore, from what was discussed above, we expect the PC to have a negative correlation with grammatical classifications pertaining to syntax, and a positive correlation with those pertaining to morphology.

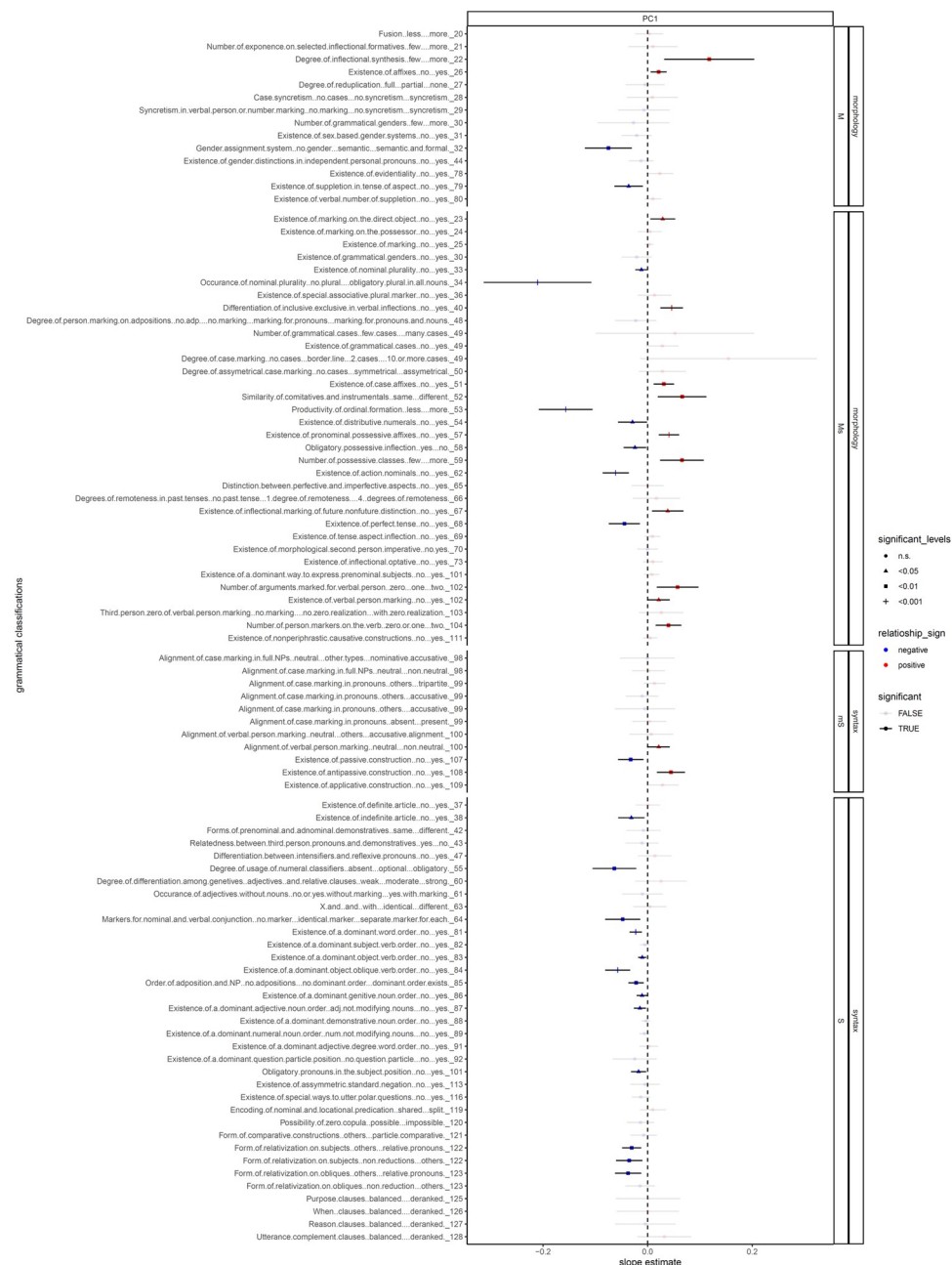

**Fig 2. The linear / logistic regression coefficients between linguistic features and societal features.** Linguistic features pertaining to complexity (y-axis) are drawn from the WALS database (Dryer & Haspelmath, 2013). The values within each feature are coded into different categories such that the complexity of each category varies from low to high. The features are faceted by 1) whether they pertain to morphology or syntax (the outer facet) and 2) whether they pertain purely to morphology or syntax, or consist of a mixture of both (the inner facet). Within each facet, the x-axis represents the principal component (PC) of societal features drawn from Ethonologue, Glottolog, and D-Place, also arranged by complexity. Each dot represents the result from a regression: a red dot indicates a positive relation between a societal PC and a linguistic feature, whereas a blue dot suggests a negative relation. The bar represents the 95% confidence interval. The transparency of the dots indicates whether the relation is significant: an opaque dot indicates a p-value smaller than 0.05, and a transparent one indicates a p-value greater than 0.05. **NOTE**: since a more complex sociopolitical feature corresponds to a more negative PC value, a blue dot hence indicates a positive correlation between linguistic complexity and sociopolitical complexity, whereas a red dot indicates a negative one.

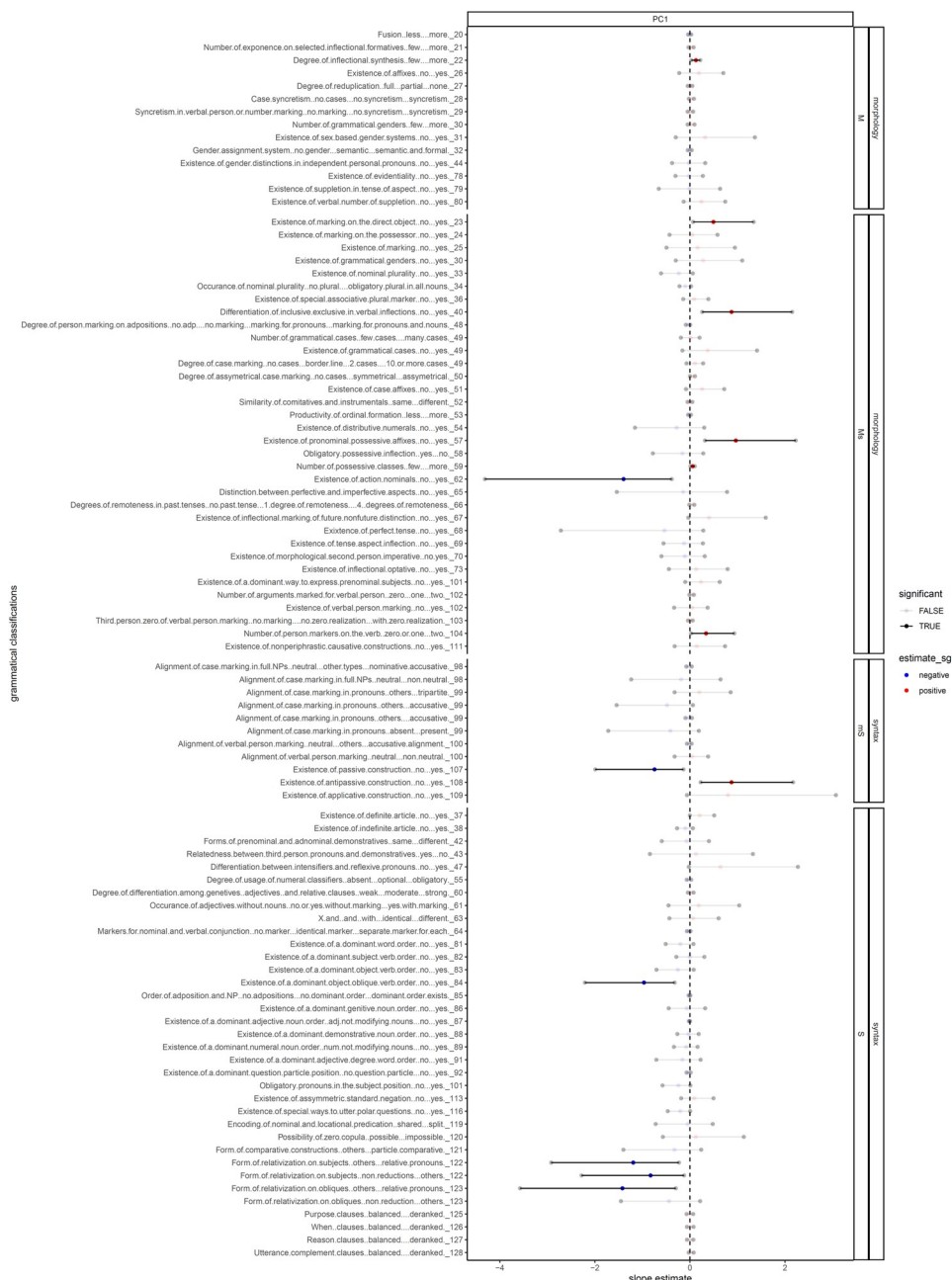

**Fig 3. The Bayesian mixed-effects linear / logistic regression coefficients between linguistic features and societal features, after controlling for language relatedness and language contact.** Linguistic features pertaining to complexity (y-axis) are drawn from the WALS database (Dryer & Haspelmath, 2013). The values within each feature are coded into different categories such that the complexity of each category varies from low to high. The features are faceted by 1) whether they pertain to morphology or syntax (the outer facet) and 2) whether they pertain purely to morphology or syntax, or consist of a mixture of both (the inner facet). Within each facet, the x-axis represents the principal components (PC) of societal features drawn from Ethonologue, Glottolog, and D-Place, also arranged by complexity. Each line segment represents the 95% credible interval of the posterior distribution of the effects of each PC on each linguistic feature. The gray dots on the edges represent the 2.5% quantile and the 97.5% quantile, respectively, and the colored dots at the center represent the posterior mean. Each dot represents the result from a regression: a red dot indicates a positive relation between a societal PC and a linguistic feature, whereas a blue dot suggests a negative relation. The transparency of the dots indicates whether the relation is significant: an opaque dot indicates significance (defined as all the 95% credible interval falls below or above zero), and a transparent dot indicates a lack thereof. **NOTE**: since a more complex sociopolitical feature corresponds to a more negative PC value, a blue dot hence indicates a positive correlation between linguistic complexity and sociopolitical complexity, whereas a red dot indicates a negative one.

In general, we found that sociopolitcal esotericity tends to correlate with morphological complexity, in the sense of more explicit markings and distinctions. From Fig 2, esotericity seems to favor more inflectional synthesis (22A), markings on the direct object (WALS Feature 23A), verbal person marking (102A), more arguments marked for verbal person (102A), more person markers on the verb (104A). Sociopolitical esotericity also correlates with case affixes (51A) and pronominal possessive affixes (57A). Finally, it seems to result in richer distinctions through more explicit markers, such as differentiating inclusive and exclusive "we" (40A), genders in independent personal pronouns (44A), comitative and instrumental "with" (52A), nouns into various possessive classes (59A), future and non-future tenses of verbs (67A), and evidentiality (78A). That said, most of these features cannot be regarded as purely morphological, since they have a syntax dimension too.

On the other hand, we also found that sociopolitical exotericity tends to correlate with more complex syntax, including more syntactic layering and more obligatory syntactic categories and distinctions. Specifically, sociopolitical exotericity favors using reduction (122A), specifically in the form of relative pronouns, to license a subject relative clause (122A) and an oblique relative clause (123A). In addition, we found that sociopolitical exotericity favors having passive constructions (107A), indefinite articles (38A), obligatory usage of numeral classifiers (55A), separate markers for nominal and verbal conjunctions (64A), and obligatory pronouns in subject positions (101A). Sociopolitical exotericity also favors obligatory word order, as we found a correlation with having a dominant word order (81A), object-verb order (83A), object-oblique-verb order (84A), adposition-NP order (85A), genitive-noun order (86A), and adjective-noun order (87A). In contrast to our findings for sociopolitical esotericity, most of the features that positively correlate to sociopolitical exotericity can be regarded as purely syntactic.

Fig 3 shows the 95% credible interval of the posterior distribution of the effect of sociopolitical complexity on each of the grammatical classifications, controlled for language relatedness and geographical proximity, and also faceted first by whether a grammatical feature is broadly considered as morphological or syntactic and then by whether a grammatical feature predominately falls into one category but has some relations with the other.

In total, 13 grammatical classifications stayed robust against controlling for the two factors. We found that sociopolitical esotericity still correlates with morphological complexity, favoring more inflectional synthesis (22A), marking on the direct object (23A), differentiating inclusive and exclusive in verbal inflections (40A), having pronominal possessive affixes (57A), more possessive classes (59A), and more person markers on the verb (104A). In addition, we found that sociopolitical exotericity still correlates with syntactic complexity, favoring having passive constructions (107A), a dominant object-oblique word order (84A), using reductions (122A) on subject relativization, and a preference for using relative pronouns to relativize subjects (122A) and obliques (123A).

Fig 4 shows the distribution of posterior means of the effect of sociopolitical complexity on each grammatical classification in the analysis after controlling for language relatedness and geographical proximity, faceted by whether these classifications pertain exclusively to morphology or syntax, or only predominantly pertain to morphology or syntax. From the figure, although only a fraction of the results are robust after language relatedness and geographical proximity are controlled, the results in the pure morphological category (M) and the pure syntactic category (S) are trending in the positive directions, in that the posterior means are mainly concentrated above zero for M and below zero for S. The results were spread between positive and negative for the two mixed categories (mS and Ms), seemingly because these classifications contain both flavors in syntax and morphology.

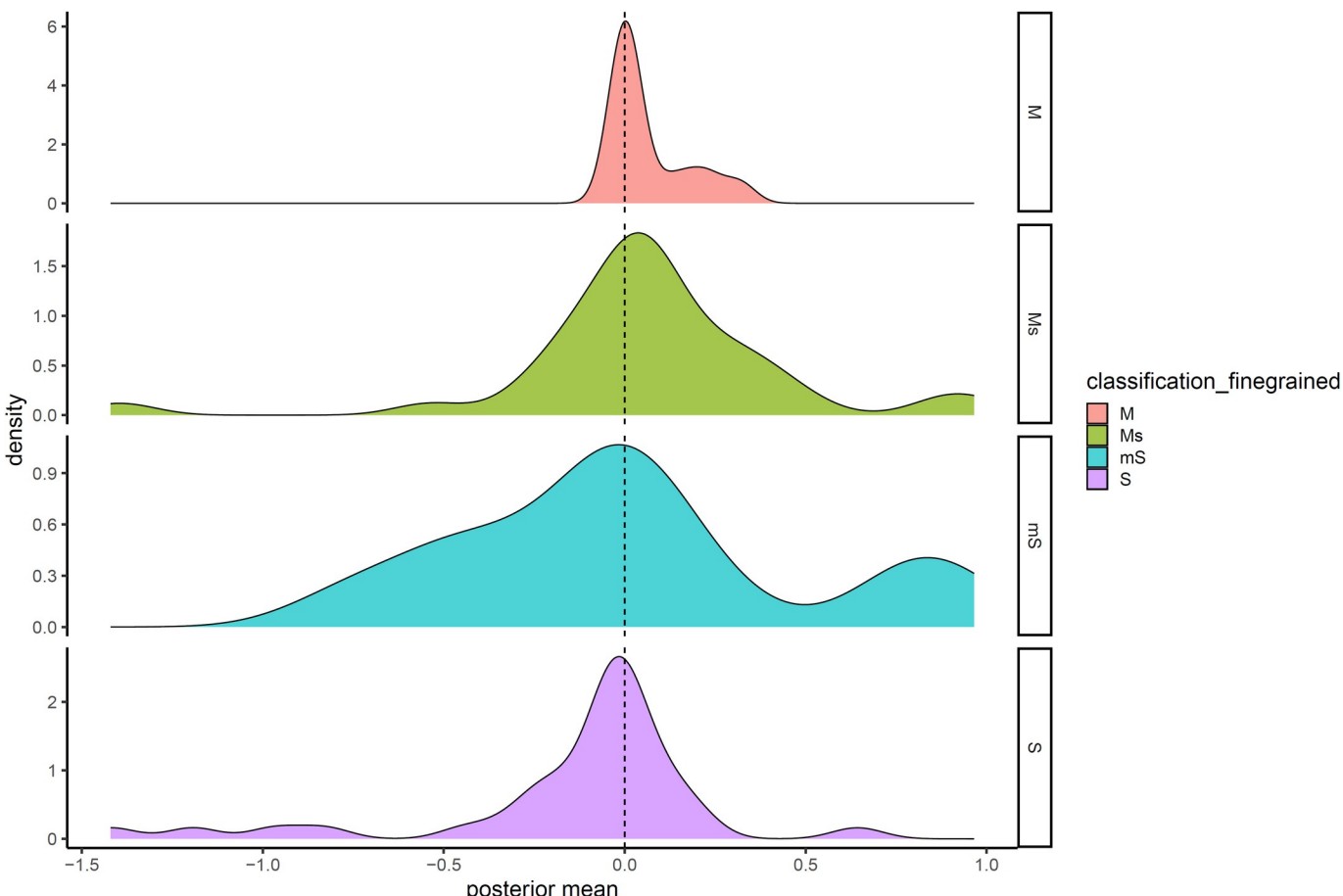

**Fig 4. The distribution of posterior means of the effects of PC on different grammatical classifications.** We obtained the posterior mean of the effect of PC on grammatical classifications in the analysis where we controlled for language relatedness and geographical proximity and plotted the distribution of these posterior means (x-axis), faceted by the fine-grained classification on each feature, namely, purely morphological (M), predominantly morphological but partially syntactic (Ms), predominantly syntactic but partially morphological (mS), and purely syntactic (S).

Meanwhile, as indicated in Figs 2 and 3, we also found a number of grammatical classifications that seem to be trending against our expectation: sociopolitical esotericity seems to favor simplicity in some morphological features, and sociopolitical exotericity seems to prefer simplicity in some syntactic ones. Specifically, we found esotericity favors less complexity in gender assignment system (32A), no suppletion in tense or aspect (33A), no nominal plurality (33A, 34A), less productivity of ordinal numerals (53A), no distributive numerals (54A), having obligatory possessive inflections (58A), and having no perfect tense (68A). On the other hand, we found exotericity favors neutral alignment in verbal person marking (100A) and no antipassive constructions (108A). Most of these features fall into the mixed categories, as they pertain to both morphology and syntax. Also, only two of these correlations (action nominals and antipassive constructions) are robust against controlling for language relatedness and geographical proximity.

## 4. Discussion

As specified in the Introduction, our overarching hypothesis for this paper is that the languages spoken by exoteric societies (Type X languages) exhibit simpler morphologies but

more complex syntaxes, the latter characterized as involving a larger number of specialized and obligatory grammatical categories and distinctions, typically implying more syntactic layering (embedding), while on the other hand, the languages spoken by esoteric societies (Type S languages) exhibit less complex/less layered syntaxes in that sense, but more complex morphologies, with more information packed into words, including more irregularity.

As reported in the Results section, our results align with our broad hypothesis. Overall, we found that Type S languages tend to exhibit more complex morphologies, when compared to Type X languages, and this seems to be the case both in nominal and verbal domains. This is particularly true for morphological features with a syntactic function, particularly for the marking of participants in the sentence through nominal or verbal inflection. At the same time, after controlling for phylogeny and geography (the two main factors accounting for language similarity), we found that only the features with a predominantly morphological function (slightly) correlate with sociopolitical simplicity. That said, a significant limitation of our study is that WALS features do not directly address the extent to which languages exhibit other typical features of Type S languages, like idiomatic expressions and formulaic language, or irregularity. Thus, our results for Type S languages are more limited than our results for Type X languages, as the parameters that pertain to syntactic complexity are well-documented in WALS, and elsewhere. One might even expect that a richer documentation of these purely morphological features (and overall, of the features typically found in Type S languages) would have strengthened the trend we have found.

By contrast, our results for syntax are consistent with the hypothesis that Type X languages are characterized by more syntactic complexity, specifically with more syntactic layering, as well as with more obligatory syntactic categories and distinctions. In particular, among the strongest findings, we observed the existence of dominant word order and obligatory pronouns in subject positions. Within the minimalist program (e.g. [53]), both relate to the syntactic layer of Tense Phrase (TP). Following this theoretical framework, the category Tense, the head of the TP, has a strong feature in some languages, requiring the specifier of TP (the subject position) to be filled, whether by moving a noun phrase from a lower layer into it, or by inserting a meaningless pronoun in this position, as found in e.g. *It is snowing* in English. This rigid rule of syntax contributes to both a dominant word order (with the subject position rigidly in the specifier of TP), and to the obligatory use of pronouns in the subject position.

Although esotericity and exotericity constitute two poles on a single scale of sociopolitical complexity (and more generally, of the effect of social organization on human communication), the factors driving the development of Type S and Type X languages might not be mirror-images but rather may be of diverse and qualitatively different natures. Thus, while the correlation between esotericity and morphological complexity could be due to factors such as simplification being due to imperfect adult second-language acquisition, the correlation between exotericity and syntactic complexification may be attributed not only to the presence of adult learners of the language, but also to factors such as the need to satisfy a broader range of communicative needs (e.g. conveying more complex meanings to unrelated people). Accordingly, for the many features associated with both morphological and syntactic complexity (those classified in Figs 2 and 3 as Ms or mS), different factors end up pulling in opposite directions. For example, for case marking (49A), a language spoken in an exoteric society might undergo reduction and loss of case-marking due to imperfect learning by adults, or alternatively develop case-marking in order to satisfy the need for greater expressive power. As a consequence, as shown in Fig 3, in this particular instance these two factors seem to cancel each other out, with no significant correlation between case marking and esotericity/exotericity. For this reason, the results of this paper, while still supporting a distinction between Type S languages with greater morphological complexity and Type X languages with greater

syntactic complexity, may not yet support the view of a trade-off between morphological and syntactic complexity (see e.g. the discussion in [17]; see also [54] for a proposal that relates these differences to differential involvement of procedural vs. declarative memories.)

In conclusion, our study is consistent with the previous findings of the existence of correlations between exotericity/esotericity and grammatical complexity, for example, [34, 55, 56]. More specifically, [56] also found an inverse correlation between exotericity and morphological complexity. On the other hand, a very recent, comprehensive study reported in [57] denies the significance of any correlations between linguistic and societal factors pertaining to esotercity/exotericity, claiming only a weak effect, and concluding in their title that "Societies of strangers do not speak grammatically simpler languages." As obvious already from this title, their study has a very different overarching hypothesis from ours, and our two studies are thus not directly comparable, even if they look at very similar phenomena, and pose very similar questions. First, their hypothesis is that any type of grammatical complexity (including morphological and syntactic) correlates inversely with societal exotericity, which is in direct opposition to our hypothesis for syntax. Given the terms they use in the paper, they test the hypothesis that languages in highly exoteric societies have (1) less phonologically fused grammatical markers (fusion) and (2) overall fewer obligatory explicit markers (informativity) compared to languages in low-exotericity societies.

Second, the syntactic parameters that we consider are much more fine-grained. Whereas the advantage of [57] is in its statistical power, our approach enables us to get more specific in identifying syntactic and morphological aspects of language variation that pertain to esotericty/exotericity, as well as to outline what further research is needed to shed light on this question. To take just one example, we consider the presence of definite and indefinite articles to be much more relevant for syntactic complexity than having a politeness distinction in pronouns, both of which are considered as equally relevant in [57] (see p.4). For example, within a minimalist approach, the presence of articles implies an additional layer of syntactic structure, such as a Determiner Phrase (DP), and it has many further ramifications for syntax beyond just the existence of two additional words. These ramifications include, but are not limited to, more rigid ordering of elements inside the DP, as well as more restrictions on the co-occurrence of different words inside a single DP, such as whether or not a possessive noun or pronoun can co-occur with a demonstrative pronoun (see e.g. [58]). In general, we believe that for studies like this it would be helpful to have more dialog between formal and typological approaches to language.

## Acknowledgments

The authors would like to express their gratitude to Russell Gray and Kaius Sinnemäki for their suggestions on improving the analysis methods, as well as to the audiences of the 56th Annual Meeting of the Societas Linguistica Europaea and the 2022 JCoLE Conference for their questions and feedback. We are especially grateful to the anonymous reviewers for their constructive feedback.

## Author Contributions

**Conceptualization:** David Gil, Ljiljana Progovac, Antonio Benítez-Burraco.

**Data curation:** Sihan Chen.

**Formal analysis:** Sihan Chen, Antonio Benítez-Burraco.

**Funding acquisition:** Antonio Benítez-Burraco.

**Investigation:** David Gil, Ljiljana Progovac, Antonio Benítez-Burraco.

**Methodology:** Sihan Chen.

**Project administration:** Antonio Benítez-Burraco.

**Supervision:** Antonio Benítez-Burraco.

**Writing – original draft:** Sihan Chen, David Gil, Sergey Gaponov, Jana Reifegerste, Tessa Yuditha, Tatiana Tatarinova, Ljiljana Progovac, Antonio Benítez-Burraco.

**Writing – review & editing:** Sihan Chen, David Gil, Sergey Gaponov, Jana Reifegerste, Tessa Yuditha, Tatiana Tatarinova, Ljiljana Progovac, Antonio Benítez-Burraco.

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
