## [Decision Letter · Decision Letter 0]

24 Aug 2023

PONE-D-23-15161Linguistic and memory correlates of societal variation: A quantitative analysisPLOS ONE

Dear Dr. Benítez Burraco,

Thank you for submitting your manuscript to PLOS ONE. After careful consideration, we feel that it has merit but does not fully meet PLOS ONE’s publication criteria as it currently stands. Therefore, we invite you to submit a revised version of the manuscript that addresses the points raised during the review process.

I am grateful to have now received thoughtful and detailed reviews from two experts in the field. Both reviewers indicate that the work has potential to make a valuable contribution and should be published after addressing some significant points in a revision. The main issue raised in both reviews is that the paper needs to brought in line with the results of the statistical tests controlling for genealogical relationship and geographical proximity.

Both reviewers also mention that the data and analyses should be made available. I know that you did make this available at my request after the submission. This was my fault for not informing the reviewers about this. (I did inform the initial reviewers, but then they did not follow through with their reviews, and I subsequently forgot to inform the new reviewers.) 

We look forward to receiving your revised manuscript.

Kind regards,

Marcus Perlman, Ph.D

Academic Editor

PLOS ONE

Journal Requirements:

"This research was supported by grant PID2020-114516GB-I00 funded by MCIN/AEI/ 10.13039/501100011033 (to ABB)."

Reviewers' comments:

Reviewer's Responses to Questions

**Comments to the Author**

1. Is the manuscript technically sound, and do the data support the conclusions?

Reviewer #1: Partly

Reviewer #2: Partly

2. Has the statistical analysis been performed appropriately and rigorously? 

Reviewer #1: I Don't Know

Reviewer #2: I Don't Know

3. Have the authors made all data underlying the findings in their manuscript fully available?

Reviewer #1: No

Reviewer #2: No

4. Is the manuscript presented in an intelligible fashion and written in standard English?

Reviewer #1: Yes

Reviewer #2: Yes

5. Review Comments to the Author

Reviewer #1: **General assessment**

The study presents the results of multiple analyses of linguistic and societal variables to establish whether the correlation between them exists and to infer if the alleged trade-off between morphological and syntactic complexity is influenced by these correlations. For this purpose, a wide range of societal variables was obtained from different databases, and the features from WALS pertaining to morphological and syntactic complexity were selected. The authors go to great lengths to overcome the Galton’s problem by rerunning the the analyses and checking if the results are supported when family relatedness and regional proximity are accounted for. The presentation of the findings and the language in general are clear and easy to follow. The authors also offer the novel interpretation of the findings in the light of the prevalence of procedural or declarative memory. Overall, I recommend the paper for publication after addressing the major concerns with 1) the presentation and interpretation of results, 2) explaining criteria underlying the distinctions between morphological and syntactic complexity and principles for including the individual WALS parameters, and 3) data availability. I also provide minor comments related to literature review and citations.

**Major comments**

**Results and their interpretation**

If my understanding is correct, only 10 tested relationships (out of 844) have survived both controlling for family relatedness and region proximity and emerged as significant (at least at 0.05 level), i.e. there are 10 cells with FR*, FR**, or FR***. These are the correlations that should be described in the results section (rather than those that additionally appear significant(**,***,***) but disappear after family or region are accounted for). While the authors make it clear which relationships hold after the additional analyses and which do not, the spurious correlations (the cells that do not contain FR*, FR**, and FR***) should not be given as much attention or interpreted as significant and showing “a clear trending in the direction predicted” (p. 15).

p. 14. “Most of the results reported in sections 3.1. and 3.2. survive controlling for family relatedness, but not all. We interpret this to mean that the correlations that do not survive can be attributed either to family relatedness or to direct correlations between societal and linguistic factors. This can also mean that both factors are contributing to the attested correlations, and that more discerning tools for testing will be needed to disentangle one 15 factor from the other. On the other hand, only very few correlations in our results survive controlling for regional factors.”

If there were direct correlations between societal and linguistic factors, the correlations would have survived controlling for family relatedness and regional proximity. The idea behind those additional tests addressing the Galton’s problem is checking whether the initial correlations hold when confounding factors are taken into account. If the correlations disappear in the analyses incorporating control for family relatedness and/or regional proximity, then these results are spurious. If the correlation remains when the effects of family and region are included in the analyses, these results cannot be explained away merely by non-independence between languages.

Additionally, I recommend to accompany the supported relationships in the results section not only with the direction of the correlation (positive/negative), but also with specifications of the strength of the correlation to clarify which relationships are weak/moderate/strong (based on the Figure 1 and the presented the correlation coefficients).

**Morphological and syntactic complexity**

The authors provide the principles based on which features pertaining to morphological complexity are included. e.g. p. 6 “We then say that the latter category is more complex than the former one, following the principle that the more symbols needed to fully describe a grammatical rule, the more complex the rule is (Li & Vitanyi, 2008). In this case, more text is needed to describe a grammar with affixes than to describe a grammar without them, since to describe the former, one needs to specify explicitly the forms, functions and locations of the affixes, whereas no description is needed for the latter.”

Do the same principles apply when syntactic complexity parameters are chosen? Or are only the criteria based on the syntactic theory (provided in the discussion) are used instead? It would be useful to provide the basic criteria for inclusion the features to the syntactic complexity metric early on.

If my understanding is correct, the selection of features on the dominant word order imply that dominant word order is treated as more complex. However, if the rules criteria is applied, a language with no dominant word order could count as more complex as more rules could be required to describe the conditions under which different orders are used. Please clarify.

Furthermore, how exactly are the distinctions made between features belonging to either morphological or syntactic complexity sets? For instance, why is the nominal plurality feature is the part of syntactic complexity, whereas future tense is classified as morphological complexity (if both features can be marked inflectionally)?

Please also clarify whether the inclusion of Accusative alignment as a parameter in syntactic complexity implies that e.g. Ergative alignment would be “less complex”.

**Data availability**

Based on the columns in the attached .csv file, the supplementary materials contain only data from Glottolog (columns on sampled languages, their location, family, etc). However, the values for WALS variables along with variables from DPlace and Ethnologue are not provided, which prevents the interested readers from replicating the study.

Furthermore, I suggest making the code used for running the analyses, data wrangling and generating the main results figure (Figure 1) available.

While based on the description of the methods, the statistical analysis appear to have been performed appropriately and rigorously, the absence of the code (for analyses) currently makes it impossible conclude whether it is the case, so I highly recommend making the code available.

**Minor comments**

p. 3 “Increasing evidence suggests, however, that overall language complexity might differ cross- linguistically (e.g., Sampson, Gil, & Trudgill, 2009; McWhorter, 2011; Koplenig, Wolfer, & Meyer, 2022).” This implies that the trade-off hypothesis is not taken for granted and many studies reveal cross-linguistic correlation rather than an expected negative correlation, but I did not encounter any explicit claims about the absence of (overarching) trade-offs that would reference the studies that find no evidence for trade-offs (examples could be Shosted (2006), Sinnemäki (2008), Miestamo (2009)).

At the same time, since the paper consistently mentions the trade-off between morphology and syntax, it would be helpful to cite studies that suggest (and ideally test) this specific trade-off.

p. 3. “Moreover, evidence also suggests that language features and the trade-offs between subparts of language grammars can be impacted by extralinguistic factors.” The works cited after this sentence involve the examination of the relationships between individual features or aggregated scores and societal features, but none of them explore how “the trade-offs between the subparts of language grammars” are influenced by extralinguistic factors. In fact, few works address how the trade-offs can be impacted by such factors. Sinnemäki (2020) could be treated as such a study, which is not cited (though Sinnemäki 2009, which is cited, explores an adjacent question).

p. 3. Please briefly spell out how exactly vegetation has an influence on sound inventories (the way it is done for tonal languages and dry climates right after the following sentence): “A familiar example is the effect of vegetation on sound inventories (Maddieson and Coupé, 2015), which parallels what has been observed in many vertebrates (Boncoraglio and Saino, 2007; Ey and Fischer, 2009)”.

p. 4. I suggest citing appropriate literature to support the claims made in this sentence: “Recent typological surveys suggest that the number of speakers, the degree of bilingualism, the tightness or the looseness of the social networks, the sociopolitical organization, or the number of adult learners of a language correlate, and perhaps explain, the types of morphology or syntax exhibited by the world languages”.

p. 4. Since Lupyan & Dale (2010) do not refer to their metric scores as “the index of agglutination”, but use “morphological complexity measure/score”, this term might be more appropriate.

p. 4. Replacing or explaining the “hypotaxis” term at first usage will make the sentence more transparent to non-specialists. Regarding the same term, it is not clear whether it is used in the same meaning as “(syntactic) layering” or if these are two different concepts: see p. 14 "a language with more layering and more hypotaxis”, p. 15 “syntactic layering (hypotaxis)”, p. 16 “syntactic layering and hypotaxis”

p. 4. Please clarify whether the distinction between Type A and Type B languages follows Kusters (2003) where a very similar distinction is made.

p. 8. The inclusion of the “permission of cousin marriages” variable needs to be motivated given that this variable has never been used or mentioned as a potential variable in the previous studies. ****

p. 9. “We consider that a correlation exists between a linguistic feature and a societal factor even if just one of the societal parameters shows such a correlation, but clearly the correlation should be considered stronger if multiple societal parameters correlate with a linguistic parameter.” Since such a diverse set of societal variables results in different language samples on which individual pairwise correlations are tested, the correlation should not be considered stronger or weaker based on how many societal parameters are correlated with linguistic variables.

**Citations and references**

p. 13 “(e.g., Bošković, 2008, and much subsequent work).” and p. 15 “(e.g., Lyons 1999, and subsequent work)”. Please specify the subsequent work.

p. 17. "In their own words, they test the hypothesis “that languages in highly exoteric societies have (1) less phonologically fused grammatical markers (fusion) and (2) overall fewer obligatory explicit markers (informativity) compared to languages in low-exotericity societies.” Please provide the page for the direct quote.

Make sure that the references of the journal articles are formatted uniformly. For instance, the journal name in Ullman (2004) and some other references is not in italics.

**References**

Kusters, Wouter. 2003. *Linguistic complexity*. Netherlands Graduate School of Linguistics.

Miestamo, Matti. 2009. Implicational hierarchies and grammatical complexity. In Sampson Geoffrey, David Gil & Peter Trudgill (eds.), *Language complexity as an evolving variable*, 80–97. Oxford: Oxford University Press.

Sinnemäki, Kaius. 2020. Linguistic system and sociolinguistic environment as competing factors in linguistic variation: A typological approach. *Journal of Historical Sociolinguistics* 6(2). 20191010. https://doi.org/doi:10.1515/jhsl-2019-1010.

Sinnemäki, Kaius. 2008. Complexity trade-offs in core argument marking. In Matti Miestamo, Kaius Sinnemäki & Fred Karlsson (eds.), *Language complexity: Typology, contact, change*, 67–88. Amsterdam/Philadelphia: John Benjamins.

Shosted, Ryan K. 2006. Correlating complexity: A typological approach. *Linguistic Typology* 10(1). 1–40.

Reviewer #2: Despite my skepticism (which I will explain below), this work should definitely be published (with minor revisions). It is well written, thought-provoking, and brings together systematic comparable data from different disciplines (linguistics, anthropology, neuropsychology), where linguistic and societal data come from large published databases. Such work should be encouraged and will certainly be a stimulus for future interdisciplinary research.

My basic problem with the article is that, in my opinion, the data presented do not sufficiently support the main hypothesis, which is: "languages spoken by exoteric societies tend towards more complex syntaxes, while languages spoken by esoteric societies tend towards more complex morphologies." (from the abstract).

Even if one accepts that syntax and morphology can be separated as clearly as the authors seem to assume (which I don't think is the case), the statistical tests would have to yield different results to confirm the initial hypothesis. If I understand the authors' method correctly, they are trying to correlate linguistic features related to morphological and syntactic complexity with socio-demographic features. If correlations emerge, the important question remains whether these correlations can also be explained by genealogical relationship ("Family") and/or by geographical proximity ("Region"). This means that all potential linguistic features must first survive controlling for family AND region before they can be interpreted as evidence for the main hypothesis above.

It looks like the authors have a completely different take on this. Because according to their quantitative evaluation, there are only very few linguistic features in Fig 1 that show a significant correlation (marked with an asterisk) and at the same time survive controlling for family and area (FR) (for instance D. Possessive Classification; E. Presence of Action Nominal Constructions; G. Use of Relative Pronouns on Subjects and Use of Relative Pronouns on Obliques: BTW, I doubt the accuracy of this last result given the clear areal distribution of feature values in the two relativization features).

And yet the text of the article pretends over and over that this lack of evidence is not a lack of evidence at all. Instead, it is said that the many linguistic features that can be explained by family and/or region are also good arguments for the hypothesis cited above. See for example this quote: "Most of the results reported in sections 3.1. and 3.2. survive controlling for family

relatedness, but not all. We interpret this to mean that the correlations that do not survive can be attributed either to family relatedness or to direct correlations between societal and linguistic factors.” (p14) But then the authors admit: "On the other hand, only very few correlations in our results survive controlling for regional factors.” (15) But nothing follows from this in their argument. Instead the authors conclude: "As reported in the Results section, our results align with our broad hypothesis.” (p16) This kind of reasoning is somehow unsatisfying for me. I would expect a revision to address this mismatch more clearly.

The declarative/procedural memory discussion should be relegated to the background or omitted altogether, because nothing is provided except the formulation of predictions: this is too weak to be published in this form. A systematic cross-linguistic data collection on formulaic, irregular language use would probably be the first step to do. But conceptually, such an endeavor is extremely tricky, since we know of vast numbers of formulaic constructions at all levels and domains of grammar in all sorts of language varieties. I wonder what the comparative concept of "formulaic use" might be, and how one might obtain meaningful data without working with language corpora (and frequencies of types/tokens). Given this complex context, it is far from clear how the projected testing of the hypothesis that different memory types are associated with different language types can be performed using "standard cognitive experiments" (p19).

Accordingly, the title of the paper should be changed because no empirical research is presented on structural linguistic features and a possible correlation with the two types of memory. This article is *only* about potential correlations between linguistic and societal features. This should also be reflected in the title.

Please provide the statistical data as well as the data on feature classification (plus the 94 WALS features, which are only retrievable from Fig a) in the supplementary data (is it there? I haven't found it). Please also highlight *which* of the societal features are involved in which positive/negative correlations. As far as I can see, this information is not given systematically, with the exception of Figure 1, where everything is squeezed in but not easily retrievable by the reader.

6. PLOS authors have the option to publish the peer review history of their article (what does this mean?). If published, this will include your full peer review and any attached files.

Reviewer #1: No

Reviewer #2: No

---

## [Author Response · Author response to Decision Letter 0]

27 Jan 2024

Our replies to reviewers can be found in the attached document

---

## [Editor Report · Decision Letter 1]

6 Mar 2024

Linguistic correlates of societal variation: A quantitative analysis

PONE-D-23-15161R1

Dear Dr. Benítez Burraco,

We’re pleased to inform you that your manuscript has been judged scientifically suitable for publication and will be formally accepted for publication once it meets all outstanding technical requirements.

Kind regards,

Marcus Perlman, Ph.D

Academic Editor

PLOS ONE

Additional Editor Comments (optional):

I have read over your revised manuscript and revision letter. I thought you did a detailed and thorough job of addressing the reviewers' comments, and I am happy to accept the manuscript for publication.
---

## [Editor Report · Acceptance letter]

3 Apr 2024

PONE-D-23-15161R1 

PLOS ONE

Dear Dr. Benítez-Burraco, 

I'm pleased to inform you that your manuscript has been deemed suitable for publication in PLOS ONE. Congratulations! Your manuscript is now being handed over to our production team.

Kind regards, 

on behalf of

Dr. Marcus Perlman 

%CORR_ED_EDITOR_ROLE%

PLOS ONE